# Prior-free Balanced Replay: Uncertainty-guided Reservoir Sampling for Long-Tailed Continual Learning

## ABSTRACT

Even in the era of large models, one of the well-known issues in continual learning (CL) is catastrophic forgetting, which is significantly challenging when the continual data stream exhibits a long-tailed distribution, termed as Long-Tailed Continual Learning (LTCL). Existing LTCL solutions generally require the label distribution of the data stream to achieve re-balance training. However, obtaining such prior information is often infeasible in real scenarios since the model should learn without pre-identifying the majority and minority classes. To this end, we propose a novel **Prior-free Balanced Replay (PBR)** framework to learn from long-tailed data stream with less forgetting. Concretely, motivated by our experimental finding that the minority classes are more likely to be forgotten due to the higher uncertainty, we newly design an uncertainty-guided reservoir sampling strategy to prioritize rehearsing minority data without using any prior information, which is based on the mutual dependence between the model and samples. Additionally, we incorporate two prior-free components to further reduce the forgetting issue: (1) Boundary constraint is to preserve uncertain boundary supporting samples for continually re-estimating task boundaries. (2) Prototype constraint is to maintain the consistency of learned class prototypes along with training. Our approach is evaluated on three standard long-tailed benchmarks, demonstrating superior performance to existing CL methods and previous SOTA LTCL approach in both task- and class-incremental learning settings, as well as ordered- and shuffled-LTCL settings.

## CCS CONCEPTS

• **Computing methodologies → Computer vision**.

## KEYWORDS

Long-Tailed Distribution, Continual Learning, Catastrophic Forgetting, Uncertainty Estimation

## 1 INTRODUCTION

Over the last decade, deep neural networks (DNNs) have demonstrated remarkable performance in various multi-media tasks, such as image segmentation [50], video caption [25], and audio-visual learning [37]. However, these tasks are usually performed in a static environment where all data is available in a single training session. In a dynamic environment where data arrives phase by phase, the

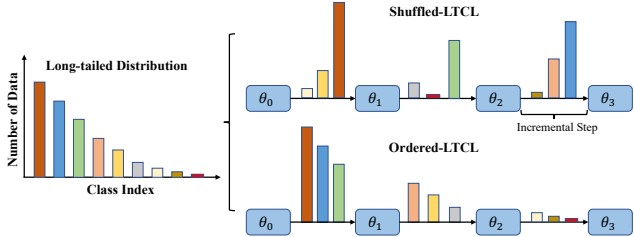

**Figure 1: Illustration of Ordered-LTCL (bottom) and Shuffled-LTCL (top).** $\theta_i$ **denotes the model parameters at the incremental step** $i$**. Ordered-LTCL assumes that all tasks are ordered by the sample number per task,** *i.e.,* **old tasks contain majority classes while new tasks have minority classes. Shuffled-LTCL assumes that all classes are randomly distributed.**

model trained on a new task tends to forget a significant amount of information of old tasks, which is commonly known as catastrophic forgetting issue [19, 27]. Besides, it was reported that recent advanced large vision and language models still face challenges in forgetting knowledge when dealing with complex sequential tasks [4], resulting in the potential loss of previously learned information during the fine-tuning process for novel tasks [14]. To overcome the catastrophic forgetting, significant efforts have been made to learn from the sequential data stream without forgetting previously acquired knowledge, which is called continual learning (CL) [58].

Despite recent significant progress in conventional CL, it is typically based on the assumption that **the training data is drawn from a balanced distribution**. However, real-world data often exhibits a long-tailed distribution [13, 53, 55], where only a few classes dominate the most samples. For instance, in autonomous driving [49, 57], anomaly accidents often occur with lower probabilities than the frequent safe events. When constructing a medical dataset [39], it is an usual phenomena that common symptoms are easily collected while it is difficult to collect enough rare symptoms. In dynamic environments characterized by realistic scenarios, minority classes often incrementally emerge as new tasks, posing a great challenge for the adaptation ability of DNNs [19]. Besides, existing CL methods exhibit severe performance degradation over the long-tailed data stream. **Therefore, it is essential to investigate continual learning over the long-tailed data stream**.

Inspired by [30], we consider two different long-tailed continual learning (LTCL) settings, *i.e.*, ordered- and shuffled-LTCL settings as shown in Figure 1. To address the LTCL problem, several straightforward solutions have been proposed [15, 21, 30] to combine existing CL methods with re-balancing techniques, such as data re-sampling [10], data re-weighting [38], and two-stage training [20]. For instance, [11] explored a balance sampling strategy to keep a balanced memory buffer for the imbalance continual learning. [36] proposed

to utilize data augmentation for the memory buffer to alleviate the class imbalance issue in the class incremental learning. **However, these approaches may not be practical for tasks evolving over time, as most re-balance techniques require label distribution information of the entire stream, while obtaining such information is often infeasible due to unknown prior of new tasks emerging in the future**. In fact, it is usually unknown whether the incoming data or the emerging class is from majority or minority classes in the real world. This makes most existing methods, which are to solve the class imbalance problem unsuited for real-world applications.

To further cast light on challenges for the LTCL problem, we conduct a motivating experiment (details can be seen in Sec 3.1) and observe that: (1) Minority samples are more likely to be forgotten than majority samples; (2) The classifier weights are easily biased to the old majority classes; and (3) Minority data is usually distributed around the task boundaries with higher uncertainty. Besides, with the limited storage and computing resource, one should utilize a constrained buffer size throughout the entire training phase to address the LTCL problem [5].

To address the LTCL issue, motivated by the above experimental findings, we propose a novel Prior-free Balanced Replay (PBR) framework to incrementally learn an evolved representation space for the LTCL problem. More precisely, we design an uncertainty-guided reservoir sampling strategy to prioritize storing minority samples in the replay memory, which is based on the mutual information between changes in model parameters and prediction results. Besides, two prior-free components are newly designed to effectively alleviate the catastrophic forgetting issue under LTCL, especially for minority classes. In detail, prototype constraint ensures all classes have balanced magnitudes by maintaining the consistency of class prototypes learned at different times, while boundary constraint prevents forgetting task boundary information by preserving boundary supporting samples of old tasks.

In summary, key contributions of this work are threefold:

**(1)** We propose a novel PBR framework to address the LTCL problem without relying on prior information (*i.e.*, label distribution), which utilizes an uncertainty-guided reservoir sampling strategy to achieve a balanced replay with less forgetting.

**(2)** We design two new prior-free components (*i.e.*, boundary and prototype constraints) to further reduce the forgetting of minority data, which can be integrated into the PBR framework based on the uncertain samples.

**(3)** Extensive experiments are conducted to evaluate the proposed method on three popular datasets under the LTCL setting, *i.e.*, CIFAR-10 [24], CIFAR-100 [24], and Tiny ImageNet [2]. Experimental results indicate that our method can achieve state-of-the-art (SOTA) performance for both task- and class-incremental learning settings, as well as ordered- and shuffled-LTCL settings, surpassing previous works by a significant margin.

## 2 RELATED WORK

***Continual Learning***. To enhance the adaption ability of deep neural networks, existing solutions could be roughly divided into four groups: rehearsal-based [40, 44, 45], distillation-based [26, 41], architecture-based [34, 48], and regularization-based methods [23].

Rehearsal-based methods [1, 5, 32] store a data subset of the old tasks and replay these samples to alleviate catastrophic forgetting. The key is to achieve effective sample selection for rehearsal, such as experience replay [40, 44] and gradient-based sample selection [1, 32]. Another solution is to imitate the previous tasks' behaviors when learning new ones. The main idea is knowledge distillation [17] taking past parameters of the model as the teacher. Besides, it is a common choice to combine rehearsal and distillation by self-distillation learning [41]. Besides, regularization-based methods mainly focus on preventing significant updates of the network parameters when learning new tasks, such as elastic weight consolidation [23], synaptic intelligence [47] and Riemannian walk [7]. Furthermore, architecture-based methods [34, 48] distinct different tasks by devoting distinguished parameter sets. For instance, Progressive Neural Networks [46] incrementally introduce a new set of parameters for incoming tasks to tackle the forgetting problem.

***Long-Tailed Learning***. To address the long-tailed problem, re-balancing strategies are the most common solutions, including re-sampling [10] and re-weighting [38]. However, these methods easily lead to performance degradation for head classes and overfitting issues for tail classes. Two-stage based methods are proposed to further improve the re-balancing strategies, such as decoupled training [20] and deferred re-balancing schedule [6]. Besides, to learn a high-quality representation space based on imbalanced data, regularization-based approaches are proposed to increase interclass differences, such as margin [6], bias [35, 42], temperature [54] or weight scale [20]. Recent works explore flexible ways for re-weighting by hard sample mining [28, 29], meta-learning [43], and influence function [38], which target to measure the importance of each training sample. Other studies propose to transfer useful knowledge from head to tail classes via designing memory module [31] or translation [22].

***Long-Tailed Continual Learning***. There are some recent works exploring the imbalance issue in continual learning, *e.g.*. Partitioning Reservoir Sampling (PRS) [21] and LT-CIL [30]. PRS [21] proposed a balance sampling strategy for head and tail classes along with the sequential tasks to preserve balanced knowledge. LT-CIL [30] utilized a learnable weight scaling layer to decouple representation learning from classifier learning. However, these methods ignore the relationship between the tasks of imbalanced and continual learning and rely on the label distribution for re-balance strategies, while our work is orthogonal with them to learn an evolved feature space for the long-tailed continual learning without label distribution. Besides, [36] proposed to utilize data augmentation for the memory buffer to alleviate the class imbalance issue, which only focused on the class imbalance in the current incremental step. [11] explored the class imbalance for online continual learning, but it ignores the unequal roles for different samples in the memory buffer.

## 3 METHODOLOGY

**Problem Formulation.** A standard learning agent sequentially observes a data stream $\{(D_0, t_0), \ldots, (D_i, t_i), \ldots, (D_{n-1}, t_{n-1})\}$, where $D_i = \{(\mathbf{x}_k^i, \mathbf{y}_k^i)\}_{k=1}^{s_i}$ is a labeled dataset of task $t_i$. $n$ is the time index indicating the task identity. $D_i$ consists of $s_i$ pairs of samples

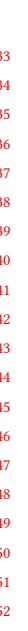
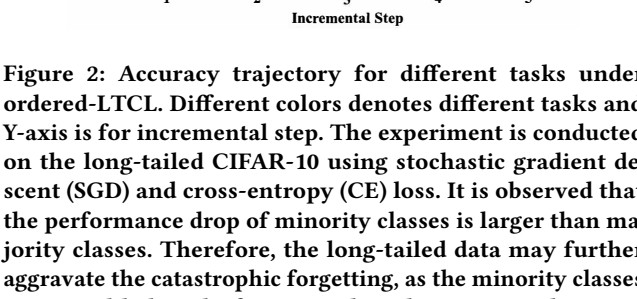

Figure 2: Accuracy trajectory for different tasks under ordered-LTCL. Different colors denotes different tasks and Y-axis is for incremental step. The experiment is conducted on the long-tailed CIFAR-10 using stochastic gradient descent (SGD) and cross-entropy (CE) loss. It is observed that the performance drop of minority classes is larger than majority classes. Therefore, the long-tailed data may further aggravate the catastrophic forgetting, as the minority classes are more likely to be forgotten than the majority classes.

with corresponding targets from the data space $\mathcal{X} \times \mathcal{Y}$. To reflect the general long-tail phenomena, we assume that the sequence $\{s_0, s_1, \ldots, s_{n-1}\}$ exhibits a power-law distribution, $i.e.$, $s_i = C\alpha^{t_i}$, where $C$ is the exponent of the power and $\alpha$ is the imbalance ratio for general long-tailed settings [31]. This assumption of power-law behavior is commonly observed in empirical distributions, and reflects how frequently samples from each task are observed.

For the classification task, the learning agent predicts the label for a given input $\mathbf{x}$ as $f_\theta(\mathbf{x})$, where $f_\theta(\cdot)$ is a mapping function from the input $\mathcal{X}$ to the output $\mathcal{Y}$ parameterized by $\theta$. Let $\ell : \mathcal{Y} \times \mathcal{Y} \to \mathbb{R}$ be the loss function between a prediction $f_\theta(\mathbf{x})$ and the target $\mathbf{y}$. Our goal is to learn the optimal parameter $\theta$ with strong continual adaptation ability to correctly classify samples from any observed tasks. The training and inference processes do not rely on task identities $t_i$. The optimization objective for the parameter $\theta$ over the data stream is given by the follows:

$$\underset{\theta}{\arg\min} \sum_{i=0}^{n-1} \mathcal{L}_{t_i}, \text{ where } \mathcal{L}_{t_i} \triangleq \underset{(\mathbf{x},\mathbf{y})\sim D_i}{\mathbb{E}} [\ell(\mathbf{y}, f_\theta(\mathbf{x}))]. \quad (1)$$

## 3.1 Motivating Experiment

As a motivation, we conduct an empirical experiment as the motivation to observe how the representations changes under the ordered-LTCL setting. We focus on two main factors in a representation space, $i.e.$, class prototypes and task boundaries. Overall, we empirically investigate the reason for the severe performance degradation of existing CL methods under the LTCL setting, $i.e.$, the deterioration of the representation space caused by biased prototypes and easily forgotten task boundaries. Here, the baseline is ResNet18 [16] trained by the SGD optimizer without any further operations ($e.g.$, memory buffer). Main results are visualized in Figure 3 and Figure 4(b), respectively.

***Forgotten Minority Data.*** As shown in Figure 2, different color denotes different tasks, $i.e.$, task-1 has the most data and task-5

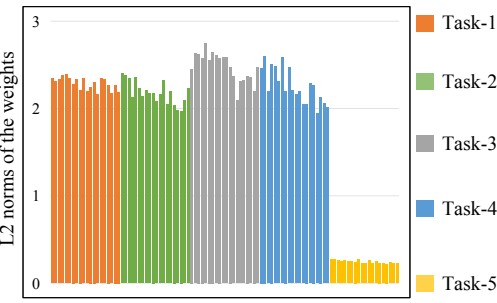

Figure 3: Biased Prototypes. The magnitudes of classifier weights are irregularly distributed due to the long-tailed continual data, producing a biased prototype for each class.

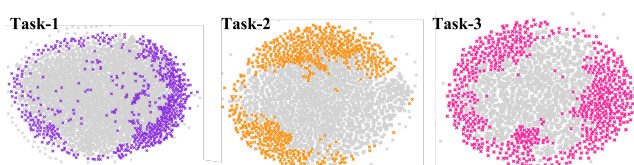

Figure 4: Forgotten Task Boundaries. We visualize the feature distribution of task $i$ at the stream end. Colorful and gray points denote the forgotten and non-forgotten samples, respectively. We found that forgotten samples contain more minority data and are generally located near the task boundary in the feature space, leading to confusing task boundaries.

has the least data, where the tendencies can reflect the forgetting degree of each task. We observe that the tendency for minority classes (yellow curve) is steeper than majority classes (blue curve), indicating that the minority classes are more likely to be forgotten than the majority classes.

***Biased Prototypes.*** As shown in Figure 3, the weight magnitudes of the classifier are irregularly distributed. The weight magnitudes of old majority classes are significantly higher than those of new minority classes, while the bias magnitudes of new classes are higher than those of the old classes. Based on such phenomena, the features of each class prototypes are generally clustered with the lack of discrimination in the representation space.

***Forgotten Task Boundaries.*** As shown in Figure 4, the boundary supporting samples of each task ($i.e.$, colorful points) are easily forgotten along with the continually arrived data, which would lead to confusing task boundaries in the representation space. In particular, the boundary supporting samples of the tasks with minority classes are more likely to be forgotten than majority classes due to insufficient training data.

## 3.2 Prior-free Balanced Replay

Motivated by the above observations, we propose a novel PBR framework based on the uncertainty-guided reservoir sampling and two constraints. The proposed approach is shown in Figure 5.

***Experience Replay.*** Experience replay aims to preserve useful knowledge of the previous tasks. Here, we explicitly store the most

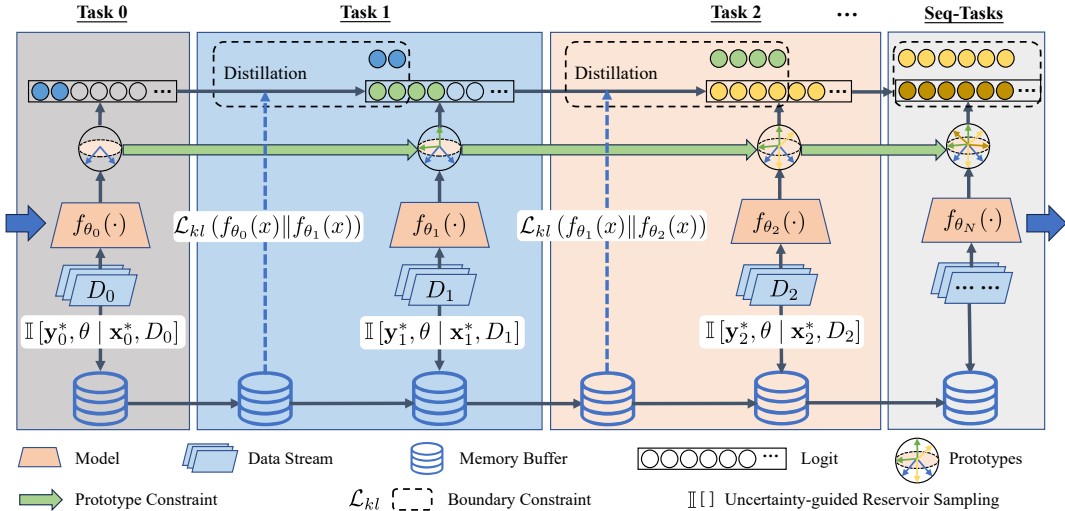

**Figure 5: The pipeline of the proposed PBR framework. It adopts a memory buffer to maintain previous experiences, where dark knowledge is distilled by teacher-student architecture. The sample selection is an uncertainty quantification problem combined with reservoir sampling. Prototype constraint could maintain the consistency of learned class prototypes for maintaining balanced predictions along with training. Boundary constraint measure the mutual dependency between the model and training samples, which can help to preserve boundary supporting samples of old tasks and maximize dissimilarities among seen tasks.**

uncertain samples of the old tasks along the incremental trajectory to improve the continual adaption ability of neural networks. To this end, we seek to minimize the following objective at the time $t_c$:

$$\mathcal{L}_{t_c} + \alpha \sum_{i=1}^{c-1} \mathbb{E}_{\mathbf{x} \sim D_i} \left[ \mathcal{L}_{kl} \left( f_{\theta_i^*}(\mathbf{x}) \| f_\theta(\mathbf{x}) \right) \right], \quad (2)$$

where $\theta_i^*$ is the optimal parameters after time $t_i$, and $\alpha$ is a hyper-parameter. $\mathcal{L}_{kl}$ is the knowledge distillation loss. To overcome the unavailability of $D_i$ from old tasks, we introduce a small memory buffer $\mathcal{M}$ to retain previous experiences. The objective seeks to replay the learned experiences by resembling the teacher-student trick. To save resources, we merely store the latest model state at $t_{c-1}$ rather than a checkpoint sequence from $t_0$ to $t_{c-1}$. In this work, we aim to maintain prototype knowledge and task boundary information by incorporating cosine normalization for different tasks. As follows, we will present the details in turn.

***Uncertainty-guided Reservoir Sampling.*** To further keep a balance between old and new tasks, we present an uncertainty-guided reservoir sampling to guarantee that uncertain samples are stored in the buffer $\mathcal{M}$ in priority, since uncertain samples are more likely to belong to the minority classes. Different from vanilla reservoir sampling [52], the uncertainty-guided sampling process is conducted at the end of each task to maintain a well-rounded knowledge. At the end of task $t_c$, the training samples $D_c$ are sorted as $\hat{D}_c$ according to their mutual dependence with the network. Then the sampling process iterates $s_c$ times between sample-in and sample-out for each $(x, y) \in D_c$. The sample-in decides whether to sample a data point into the memory, while the sample-out removes a sample from the memory.

**(1) Candidate:** To store uncertain samples, for each iteration, we first generate a candidate sample by:

$$\mathcal{K} = \{(\mathbf{x}^*, \mathbf{y}^*) \mid (\mathbf{x}^*, \mathbf{y}^*) = \underset{(\mathbf{x},\mathbf{y}) \in \mathcal{D}_c}{\arg\min} \left( \mathbb{I}\left[\mathbf{y}, \theta \mid \mathbf{x}, D_c\right] \right)\}. \quad (3)$$

where $\mathbb{I}[\cdot|\cdot]$ indicates the mutual information (MI) between the prediction and the posterior over parameters. If $\mathcal{K}$ is already stored in the memory, we will generate a new candidate from $D_c \backslash \{K\}$, otherwise keeping it fixed.

**(2) Sample-In:** We design a probability function $\mathcal{P}(\mathcal{K})$ to decide whether moving $\mathcal{K}$ into the memory $\mathcal{M}$:

$$\mathcal{P}(\mathcal{K}) = \frac{|\mathcal{M}|}{N_c + \sum_i^{c-1} s_i w_i}, \quad \text{where } w_i = \frac{e^{-s_i}}{\sum_{j=1}^{c-1} e^{-s_j}}, \quad (4)$$

where $N_c$ is the total sampling iteration number from the end of task $t_c$ up to now. $s_i$ is the running frequency of class $i$. This condition implicitly achieves a trade-off between old and new tasks.

**(3) Sample-Out:** If the memory is out of buffer size, a sample will be removed when a candidate is entered into the memory $\mathcal{M}$. The probability that an sample is removed follows the uniform distribution over the memory size, *i.e.*, $\frac{1}{|\mathcal{M}|}$, since sample-in works towards achieving a balanced partition for old tasks.

Note that due to the lack of sufficient knowledge about the minority classes, the uncertainty degree of minority data is generally higher. Therefore, our method implicitly encodes the rule to preferentially store minority samples, rather than directly adjusting the label distribution to alleviate data imbalance.

***Uncertainty-guided Mutual Information.*** Given a network with the limited capacity, uncertainty can be utilized to estimate the importance of each training sample and identify easily forgotten minority samples with boundary information. In this work, we

utilize Monte-Carlo dropout [12] for uncertainty quantification, which interprets dropout regularization as a variational bayesian approximation. Conditioning on a neural network on finite random variables, when training with data $D$, we could calculate the posterior probability distribution $p\left(y_t^* \mid x_t^*, D\right)$ by marginalizing out the posterior distribution, which can be approximated using Monte Carlo integration with dropout regularization:

$$p\left(\mathbf{y}_t^* \mid \mathbf{x}_t^*, D\right) \approx \frac{1}{T} \sum_{s=1}^{T} \text{softmax}\left(\mathbf{w}^T f_{\theta^s}^n(\mathbf{x}_t^*)/\tau_1\right), \quad (5)$$

which is achieved by $T$ sampled dropout masks and the masked parameters. This formulation implicitly endues the network with the ability to quantify the confidence of predicted results. Here, we employ the predictive entropy as an indication of the amount of information in the prediction distribution of the network:

$$\mathbb{H}\left[\mathbf{y}_t^* \mid \mathbf{x}_t^*, D\right] = -\sum_i^c p\left(\mathbf{y}_t^* = i \mid \mathbf{x}_t^*, D\right) \log p\left(\mathbf{y}_t^* = i \mid \mathbf{x}_t^*, D\right).$$
$$(6)$$

To capture the trustworthiness of the network in training samples, we compute the mutual information (MI) between the prediction and the posterior over parameters as the mutual dependence:

$$\mathbb{I}\left[\mathbf{y}_t^*, \theta \mid \mathbf{x}_t^*, D\right] = \mathbb{H}\left[\mathbf{y}_t^* \mid \mathbf{x}_t^*, D\right] - \mathbb{E}_{p(\theta|D)}\left[\mathbb{H}\left[\mathbf{y}_t^* \mid \mathbf{x}_t^*, \theta\right]\right]. \quad (7)$$

Based on mutual dependence, minority data with boundary information of previous tasks are stored in the buffer via an efficient way, maintaining the previously learned knowledge and maximize dissimilarities among all seen tasks.

## 3.3 Prior-free Components

Based on the selected samples in the memory, we propose two new prior-free components to further alleviate the forgetting issue.

***Prototype Constraint.*** Prototype constraint is to maintain the consistency of learned class prototypes along with training. A typical classifier produces the predicted probability of a sample $\mathbf{x}$ by:

$$p_i(x) = \frac{\exp\left(\mathbf{w}_y^T f_\theta(\mathbf{x}) + b_y\right)}{\sum_{i \in \mathcal{Y}} \exp\left(\mathbf{w}_i^T f_\theta(\mathbf{x}) + b_i\right)} \quad (8)$$

where $\mathbf{w}_i$ is the $i$-th weight vector and $b_i$ is the $i$-th bias term in the classifier. As shown in Figure 3, the weight magnitudes are irregularly distributed, resulting in biased prototypes in the feature space. To address this issue, we propose utilize two types of statistical information *i.e.*, class prototype and cosine similarity to preserve useful class-wise information.

Concretely, given an input data point $\mathbf{x}$, the mapping function $f_\theta(\mathbf{x})$ is to map $\mathbf{x}$ as a hidden representation before the final linear projection for classification. Inspired by cosine normalization [18, 33, 51], we utilize a scaled cosine classifier to extract normalized embeddings of samples by $f_\theta^n(\mathbf{x}) = \frac{f_\theta}{\|f_\theta\|}$, which produces the predicted probability as follows:

$$p_i(x) = \frac{\exp\left(s\hat{\mathbf{w}}_y^T f_\theta^n(\mathbf{x})\right)}{\sum_{i \in \mathcal{Y}} \exp\left(s\hat{\mathbf{w}}_i^T f_\theta^n(\mathbf{x})\right)}, \quad (9)$$

where $\hat{\mathbf{w}}$ denotes the normalized weights in the classifier and $\mathbf{s}$ is the scaling factor. Instead of computing the average feature over all

samples, this formulation allows us to interpret the weight vectors of the classifier as class prototypes during training, which could save the costs for computing average features. It is also noteworthy to preserve cosine similarity scores among previously learned class prototypes. Thus, we further enforce the newly updated classifier to mimic the behavior of previously learned classifier, which could produce approximately consistent similarity scores for each task along with newly coming data. Formally, we propose to exploit a distillation loss to preserve prototype information as follows:

$$\mathcal{L}_{d_c} = \sum_{i=1}^{c-1} \left\| \hat{\mathbf{w}}_{i,} - \hat{\mathbf{w}}_{i,}^* \right\|, \quad (10)$$

where $\hat{\mathbf{w}}_{i,}$ is the weight for previous task $i$ in the prototype-based classifier. Different from previous cosine normalization encouraging the similar angles between the features and the class prototypes [18], such a distillation loss enhances the learned prototypes to be approximately preserved in the current model.

***Boundary Constraint.*** Boundary constraint is to preserve uncertain samples with boundary supporting information for continually re-estimating task boundaries. Denote the incoming data by $\mathbf{X}^{in}$ and data stored in the memory buffer by $\mathbf{X}^{bf}$, we use a modified cross-entropy (MCE) loss to link prototypes and logits:

$$\mathcal{L}_{t_c}(\mathbf{x}) = -\sum_{\mathbf{x} \in \mathbf{X}^{in} \cup \mathbf{X}^{bf}} \log \frac{\exp\left(s\hat{\mathbf{w}}_y^T f_\theta^n(\mathbf{x})/\tau_1\right)}{\sum_{i \in \mathcal{Y}} \exp\left(s\hat{\mathbf{w}}_i^T f_\theta^n(\mathbf{x})/\tau_1\right)}, \quad (11)$$

where $\hat{\mathbf{w}}_i$ is $i$-th weight vector (prototype) of the classifier and $\tau_1$ is a scaling factor. The prototype is normalized so that $\hat{\mathbf{w}}_i^T f_\theta^n$ is a cosine similarity metric. Note that class prototypes are explicitly updated where samples of the same class lie close by each other. Beyond to the prototypes, we further consider uncertain samples to preserve effective boundary information via the knowledge distillation loss:

$$\mathcal{L}_{kl}\left(f_{\theta^*}(x) \| f_\theta(x)\right) =$$
$$-\sum_{i=1}^{c-1} \frac{\exp\left(s\hat{\mathbf{w}}_i^T f_{\theta^*}^n(\mathbf{x})/\tau_2\right)}{\sum_{j \in \mathcal{Y}} \exp\left(s\hat{\mathbf{w}}_j^T f_{\theta^*}^n(\mathbf{x})/\tau_2\right)} \log \frac{\exp\left(s\hat{\mathbf{w}}_i^T f_\theta^n(\mathbf{x})/\tau_2\right)}{\sum_{j \in \mathcal{Y}} \exp\left(s\hat{\mathbf{w}}_j^T f_\theta^n(\mathbf{x})/\tau_2\right)},$$
$$(12)$$

where $\theta^*$ is the parameters after the task $t_{c-1}$. $\tau_2$ is the temperature scale. We can rewrite Equation 2 as follows:

$$\mathbb{E}_{\mathbf{x} \sim \{D_c \cup \mathcal{M}\}} \mathcal{L}_{t_c}(\mathbf{x}) + \alpha \mathbb{E}_{\mathbf{x} \sim \mathcal{M}} \left[\mathcal{L}_{kl}\left(f_{\theta^*}(\mathbf{x}) \| f_\theta(\mathbf{x})\right)\right] + \beta \mathcal{L}_{d_c}. \quad (13)$$

We approximate the expectation on batches sampled from the current task and the buffer, respectively.

## 4 EXPERIMENTS

### 4.1 Experimental Setup

*4.1.1 Benchmarks.* We consider two common CL scenarios [5]: (1) Task Incremental Learning (Task-IL), where task identities are provided to select the relevant classifier for each sample during evaluation; (2) Class Incremental Learning (Class-IL), where task identities are not provided during evaluation. This difference makes Task-IL and Class-IL the easiest and hardest scenarios. We present three benchmarks for LTCL: **Seq-CIFAR-10-LT, Seq-CIFAR-100-LT, and Seq-TinyImageNet-LT**, where the training data yields

**Table 1: Accuracy for previous SOTA method on Seq-CIFAR-100-LT under the ordered-LTCL setting. IR is 0.01.**

| Method | Class-IL | Task-IL |
|--------|----------|---------|
| PODNET+ [30] | 23.90 | 46.00 |
| **Ours** | **25.05** | **52.23** |

standard long-tailed distribution as defined by [31]. Besides, motivated by [30], we also consider order- and shuffled-LTCL settings.

*4.1.2 Implementation Details.* For a fair comparison, we try our best to maintain the experimental setting as consistent as possible. We implement our framework with Pytorch and Torchvision libraries and use NVIDIA TITAN 2080 Ti GPU to train the deep neural network. Following [5], we employ ResNet18 as the basic backbone for all methods, and all networks are randomly initialized rather than pre-trained. We used stochastic gradient descent with a batch size of 32 and a learning rate of 0.03. $s$ is assigned as 10. $\tau_1$ is set as 0.1 and $\tau_2$ is 2. It is important that no pre-trained model is used in all our experiments. As for the testing phase, we utilize 128 as the batch size for validation.

*4.1.3 Comparison Methods.* **Baselines.** SGD-LT is using SGD under long-tailed distribution. SGD-BL is under balanced distribution. Both SGD-LT and SGD-BL are under the CL setting. JOINT-LT is to jointly train all tasks under long-tailed distribution, and JOINT-BL is to jointly train all tasks under balanced distribution.

**CL Methods.** To provide fair comparisons, we compare 14 models involving regularization, distillation, architecture and rehearsal based methods. Regularization-based methods include Elastic Weight Consolidation (oEWC) [23] and Synaptic Intelligence (SI) [56]. Two methods leveraging knowledge distillation are Incremental Classifier and Representation Learning (iCaRL) [41] and Learning Without Forgetting (LwF) [26]. One architectural method is called Progressive Neural Networks (PNN) [46] and eight rehearsal-based methods include Experience Replay (ER) [40, 44], Gradient Episodic Memory (GEM) [32], Averaged-GEM (A-GEM) [9], Gradient based Sample Selection (GSS) [1], Function Distance Regularization (FDR) [3], Hindsight Anchor Learning (HAL) [8], Dark Experience Replay [5], and DER++ [5]). (3) Conventional long-tailed algorithms are unavailable due to compatibility with the scenarios.

**LTCL Methods.** We also include the previous SOTA method PODNET+ [30] as the comparison, which adopts a two-stage strategy to learn a balance classifier for different tasks.

*4.1.4 Evaluation Protocols.* We adopt two evaluation metrics. The first is the average accuracy (ACC) at the end of all tasks. We run 5 times experiments and report the mean accuracy and the standard deviation. The second is backward transfer (BWT), indicating how much a new task influences the performance of previous tasks. Namely, higher negative values for BWT suggest catastrophic forgetting. ACC and BWT are calculated as follows:

$$\text{ACC} = \frac{1}{T}\sum_{i=1}^{T} R_{T,i}, \text{BWT} = \frac{1}{T-1}\sum_{i=1}^{T-1} R_{T,i} - R_{i,i}, \quad (14)$$

where $R_{j,i}$ is the accuracy for task $i$ at the end of task $j$ in the sequence, and T is the total number of tasks.

## 4.2 Comparison Results

***Comparisons with SOTA LTCL Method under Ordered-LTCL.***
As seen in Table 1, compared with previous SOTA method, our method can outperform PODNET+ [30] by a large margin for both Class-IL and Task-IL settings. The main reason is that our method can focus on uncertain samples to achieve imbalanced learning and reduce forgetting.

***Comparisons with CL methods under Ordered-LTCL.*** As shown in Table 2, our method could achieve SOTA results compared with CL algorithms. PNN achieves the best results among non-rehearsal methods but attains lower accuracy than rehearsal-based ones. Most rehearsal-based approaches achieve higher results than regularization-based ones because of replaying old samples and learning new samples together. Distillation-based rehearsal methods aim to output similar logits for old samples when learning for new tasks, but the logit information is biased due to long-tailed distribution. Additionally, the methods resorting to gradients (GEM, A-GEM, GSS) seem less effective under this setting, since class imbalance negatively impacts the gradient. Although existing CL methods could reasonably deal with the forgetting issue, they still perform poorly under long-tailed distribution.

***Comparisons under Shuffled-LTCL.*** We report the results on shuffle-LTCL scenario (Seq-CIFAR-100-LT with 5-task setting). The buffer size is 200. Due to the superior performance of the strong baseline DER++ on the continual learning task, we utilize DER++ as the baseline. The proposed method outperforms DER++ by a large margin for both Class-IL and Task-IL settings. With varying IRs, our method can still improve the final performance.

***Comparisons with Different Reservoir Sampling Strategies.***
To verify the effectiveness of uncertainty-guided reservoir sampling, we report the comparison results of our method with random reservoir sampling under ordered-LTCL. As indicated in Table 2, our method with random reservoir sampling can even outperform previous CL methods by a large margin. Furthermore, the proposed method with random reservoir sampling performs slightly lower accuracy than that with uncertainty-guided reservoir sampling, demonstrating the effectiveness of uncertainty estimation.

## 4.3 Data and Task Analysis

***Effect of LTCL..*** Table 2 reports the average accuracy results at the end of all tasks under the ordered-LTCL setting. We observe that the task-IL accuracy of SGD-LT is better than SGD-BL. The main reason is that new tasks with minority samples could reduce forgetting of old tasks with majority samples. Furthermore, since alleviating forgetting may induce imbalanced impacts on the new tasks, some approaches exhibit lower accuracy than SGD-LT. For instance, regularization-based methods (*e.g.*, oEWC, Lwf, PNN, and GEM) suffer from the imbalance issue on the new tasks, which arises from the learned regularization information from the old task. Therefore, by considering the motivating experiment in Section 3.1, our method can well address the LTCL issue by adopting prototype and boundary constraints.

***Effect of Imbalance Ratio.*** We evaluate the comparison methods with different imbalanced ratios following [31]. As shown in Table 3 and Appendix, our approach remarkably outperforms previous SOTA method DER++, where more detailed results are shown in

**Table 2: Comparisons under the ordered-LTCL setting. IR is 0.01. '-' indicates unachievable results due to compatibility issues.**

| Buffer | Method | Seq-CIFAR-10-LT | | Seq-CIFAR-100-LT | | Seq-TinyImageNet-LT | |
|---|---|---|---|---|---|---|---|
| | | Class-IL | Task-IL | Class-IL | Task-IL | Class-IL | Task-IL |
| - | JOINT-BL | 92.20 | - | 75.79 | - | 61.96 | - |
| | JOINT-LT | 70.36 | - | 61.68 | - | 33.81 | - |
| | SGD-BL | 19.66 | 61.02 | 6.56 | 17.83 | 7.92 | 18.31 |
| | SGD-LT | 19.62 | 72.65 | 6.27 | 14.13 | 1.73 | 10.81 |
| - | oEWC [23] | 17.53±0.47 | 62.26±4.52 | 8.09±0.30 | 18.23±3.22 | 0.40±0.16 | 5.08±0.97 |
| | SI [56] | 16.95±0.33 | 62.48±4.51 | 7.85±0.52 | 18.23±4.17 | 2.47±0.75 | 8.07±1.00 |
| | LwF [26] | 16.70±0.20 | 59.44±1.32 | 8.47±0.20 | 14.80±2.13 | 3.02±0.36 | 9.10±0.87 |
| | PNN [46] | - | 84.72±0.88 | - | 48.89±0.86 | - | 15.61±0.96 |
| 200 | ER [40, 44] | 39.14±1.68 | 85.72±1.01 | 14.72±2.31 | 42.32±1.17 | 5.58±0.57 | 35.31±0.67 |
| | GEM [32] | 29.20±0.97 | 82.83±0.87 | 16.15±1.02 | 43.54±1.17 | - | - |
| | A-GEM [9] | 27.00±0.67 | 77.56±1.42 | 11.91±0.45 | 30.57±1.57 | 3.34±0.24 | 12.16±0.21 |
| | A-GEM-R [9] | 17.86±0.87 | 71.44±1.64 | 6.59±0.40 | 21.62±1.57 | 2.60±0.27 | 11.02±0.29 |
| | iCaRL [41] | 40.27±3.24 | 86.90±2.12 | 22.85±3.54 | 47.62±2.21 | 7.01±0.62 | 29.13±1.68 |
| | FDR [3] | 28.54±3.17 | 72.52±0.99 | 19.63±3.01 | 46.03±0.97 | 5.98±0.34 | 31.15±0.77 |
| | GSS [1] | 35.71±3.64 | 85.74±1.87 | 11.78±4.12 | 40.56±1.57 | - | - |
| | HAL [8] | 30.91±2.97 | 75.03±2.12 | 5.18±3.20 | 17.25±2.74 | - | - |
| | DER [5] | 28.00±1.81 | 74.75±1.44 | 18.31±2.03 | 47.81±1.27 | 6.56±0.82 | 35.29±0.78 |
| | DER++ [5] | 37.65±1.87 | 83.25±1.24 | 19.62±1.95 | 46.22±1.14 | 6.92±1.53 | 34.52±1.36 |
| | **Ours - Random** | 44.77±1.23 | 82.50±1.45 | 20.17±1.78 | 48.32±1.07 | 8.06±0.95 | 34.98±1.11 |
| | **Ours - Uncertainty** | **51.00±1.74** | **89.99±1.32** | **24.18±1.67** | **50.66±0.85** | **10.24±1.44** | **36.01±1.32** |
| 500 | ER | 56.01±0.61 | 87.42±0.54 | 20.50±0.36 | 49.63±0.31 | 8.49±0.59 | 39.61±0.94 |
| | GEM | 28.46±1.77 | 84.16±0.75 | 22.47±2.45 | 49.63±1.01 | - | - |
| | A-GEM | 18.09±0.71 | 79.27±1.55 | 8.68±2.57 | 49.63±2.23 | 3.09±0.44 | 10.35±0.77 |
| | A-GEM-R | 11.32±0.67 | 57.97±1.54 | 6.70±2.46 | 23.03±2.07 | 2.42±0.43 | 11.11±0.78 |
| | iCaRL | 45.08±3.99 | 87.53±3.01 | 24.51±2.03 | 50.74±1.08 | 10.51±1.76 | 11.87±3.01 |
| | FDR | 30.76±3.58 | 84.11±1.01 | 24.99±0.54 | 50.37±0.86 | 10.21±0.45 | 39.42±1.03 |
| | GSS | 47.63±4.15 | 87.65±1.32 | 14.74±3.97 | 44.82±0.95 | - | - |
| | HAL | 39.18±4.44 | 82.50±2.36 | 14.74±4.56 | 44.82±3.19 | - | - |
| | DER | 42.25±1.94 | 86.69±0.62 | 20.47±1.01 | 50.23±1.23 | 9.22±1.23 | 43.38±1.08 |
| | DER++ | 48.68±1.88 | 88.42±0.64 | 20.90±0.98 | 51.88±1.14 | 10.42±1.47 | 41.94±1.23 |
| | **Ours - Random** | 50.73±1.53 | 86.50±1.24 | 20.94±0.97 | 50.98±1.02 | 9.86±1.32 | 41.26±1.29 |
| | **Ours - Uncertainty** | **54.61±1.87** | **91.03±1.11** | **25.05±1.32** | **52.23±1.29** | **10.78±1.25** | **44.13±1.36** |

**Table 3: Accuracy for CL approaches on standard long-tailed benchmarks under the shuffled-LTCL setting.**

| Buffer | Method | Class-IL | | | |
|---|---|---|---|---|---|
| | | 0.01 | 0.02 | 0.05 | 0.1 |
| 200 | DER++ | 35.43 | 40.35 | 45.64 | 55.03 |
| | **Ours** | **38.49** | **48.64** | **49.75** | **59.33** |

| Buffer | Method | Task-IL | | | |
|---|---|---|---|---|---|
| | | 0.01 | 0.02 | 0.05 | 0.1 |
| 200 | DER++ | 74.10 | 76.09 | 84.65 | 84.77 |
| | **Ours** | **80.42** | **83.08** | **85.72** | **87.99** |

Appendix. It is observed that the performance of existing methods is improved along with increasing imbalance ratio, which mainly benefits from the increasing sample number for each task. We notice that the imbalance ratio slightly impacts the final performance

of these methods. Thus, when adopting boundary supporting samples in the buffer, the gain from increasing the imbalance ratio for our method is much larger than other methods.

**Effect of Buffer Size.** As shown in Table 2, our method outperforms other rehearsal-based methods remarkably using different buffer sizes. The general rule is that a larger buffer size benefits the final performance. Besides, GEM, A-GEM-R, and iCaRL exhibit higher sensitivity to the buffer size, indicating that these methods are easily influenced by the random samples stored in the buffer.

**Evolution of Task.** To further justify the effectiveness of our method, we show the class-IL accuracy curves along with sequential tasks in different datasets under the ordered-LTCL setting. As shown in Figure 6, the overall accuracy tendency is downwards along with incoming new tasks due to forgetting. It can be seen that our method performs the best accuracy at the end of each task, while other approaches forget a lot with incoming new tasks.

**Ordered-LTCL Accuracy for Tasks with Minority Classes.** To indicate the performance improvement of our method on the minority data, we report the class-IL accuracy for the tasks with minority

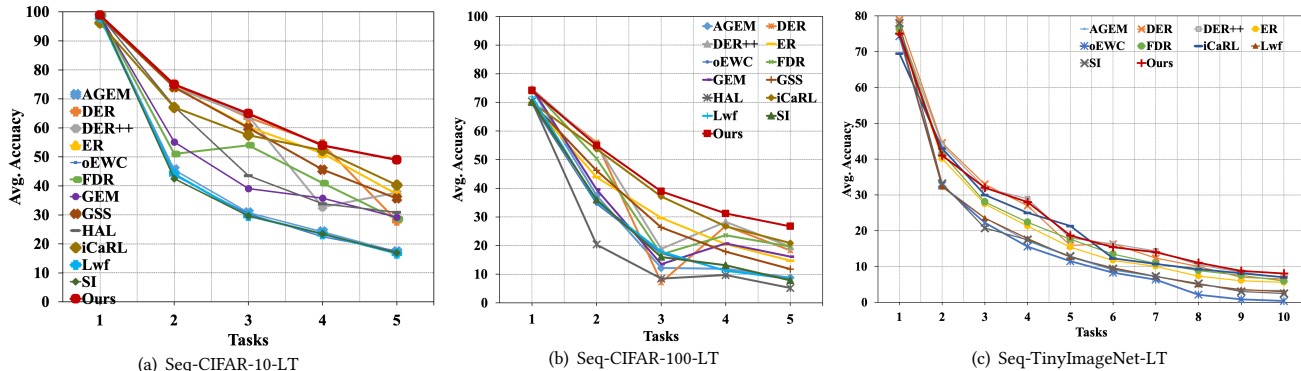

**Figure 6: Accuracy during task evolution under ordered-LTCL setting on three benchmarks. The imbalance ratio is** 0.01**. The red one denotes the best result. It is observed that our method can obtain the best performance with newly coming data and tasks.**

**Table 4: Ordered-LTCL Class-IL accuracy on Seq-CIFAR-10-LT for minority classes. IR is 0.01 and buffer size is 200.**

| Method | Task-3 | Task-4 | Task-5 |
|--------|--------|--------|--------|
| SGD    | 28.23  | 29.76  | 37.20  |
| **Ours** | 48.75 | 46.52 | 58.66 |

**Table 5: Ordered-LTCL accuracy results on Seq-CIFAR-10-LT for ablation studies. IR is 0.01 and buffer size is 200.**

| Buffer | Classifier | Class-IL | Task-IL |
|--------|-----------|----------|---------|
| - | linear | 19.27 | 69.49 |
| - | cosine | 30.24 | 74.66 |
| random | linear | 33.21 | 76.64 |
| random | cosine | 44.77 | 82.50 |
| uncertainty | linear | 33.66 | 76.45 |
| uncertainty | cosine | **51.00** | **89.99** |

classes (*i.e.*, Task-3, Task-4, and Task-5 on Seq-CIFAR-10-LT). As shown in Table 4, our method can obtain a significant performance improvement on the tasks with minority classes, while the baseline (*i.e.*, SGD) performs worse due to catastrophic forgetting.

***BWT Comparisons under Ordered-LTCL.*** BWT computes the difference between the current accuracy and its best value for each task. Lower negative values of BWT indicate that new tasks lead to more catastrophic forgetting of the previous tasks. As shown in Appendix, previous methods still forget a lot with lower negative values of BWT, while our method maximizes BWT with minimal forgetting. Our method performs significantly better than other CL methods like iCaRL, DER, and ER. The main reason is that our method can learn an well evolved feature space based on the prototypes and boundary supporting samples.

### 4.4 Ablation Analysis

***Importance of Prototype-based Classifier.*** We analyze different components of our method to verify their effects. The linear classifier is a fully-connected layer with the bias, and the cosine classifier is a normalized fully-connected layer without the bias. As shown in Table 5, our method could obtain the best results using uncertainty quantification and cosine classifier. The linear classifier without rehearsal yields the worst performance because of both catastrophic forgetting and imbalance. As the cosine classifier retains the prototype and similarity information among classes, the class-IL accuracy can be improved compared to the linear classifier. The linear classifier with uncertainty performs lower accuracy results without prototype information, although uncertainty is used to select boundary supporting samples.

***Importance of Boundary-supporting Sample.*** In this part, we analyze the effect of boundary-supporting samples (uncertainty) on

reservoir sampling. Table 5 reports the results of random reservoir sampling and uncertainty-guided reservoir sampling. It is observed that the performance is significantly reduced when using random reservoir sampling. The main reason is a lack of important boundary information due to catastrophic forgetting, although prototypes can be maintained via knowledge distillation over random samples. Based on the uncertainty estimation, the decision boundaries between old and new classes can be well modeled via replaying boundary supporting samples.

**More experimental analysis can refer to Appendix.**

## 5 CONCLUSION

In this work, we propose a novel Prior-free Balanced Replay (PBR) framework based on the newly designed uncertainty-guided reservoir sampling strategy, which prioritizes rehearsing minority data without using prior information. Additionally, we incorporate two other prior-free components to further reduce the forgetting issue including prototype and boundary constraints, which can maintain effective feature information for continually re-estimating task boundaries and prototypes. Compared with existing CL methods and SOTA LTCL approach, the experimental results on three standard long-tailed benchmarks demonstrate the superior performance of the proposed method in both task and class incremental learning settings, as well as ordered- and shuffled-LTCL settings. We believe this would be an important step towards real-world scenarios by combining continual learning with long-tailed distribution.

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
