# OpenReview forum: "Prior-free Balanced Replay: Uncertainty-guided Reservoir Sampling for Long-Tailed Continual Learning"
_acmmm.org/ACMMM/2024/Conference — MM2024 Poster_

### Official Review · Reviewer_xh48 · 2024-05-24

**Rating:** 3
**Confidence:** 2

**Summary:**

In continual learning (CL), catastrophic forgetting is a major issue, especially with long-tailed data distributions (LTCL). Traditional methods need label distribution for re-balancing, which is impractical in real scenarios. The authors propose a Prior-free Balanced Replay (PBR) framework that uses an uncertainty-guided reservoir sampling strategy to prioritize minority data without prior information. The framework also includes boundary and prototype constraints to further reduce forgetting.

**Strengths:**

1. Catastrophic forgetting is a practical and noteworthy problem.
2. The proposed method seems to show a significant improvement compared to the baseline algorithms discussed in the paper.

**Limitations:**

1. In Figure 1, the long-tail distribution appears to refer to the distribution of classes, while in the Problem Formulation, the long-tail distribution seems to pertain to the number of samples for each task. This discrepancy may appear inconsistent.

2. What is the rationale behind requiring the sample counts for each task to be arranged in descending order? This assumption appears excessively restrictive.

3. What is the rationale behind assuming that the long-tail distribution adheres to an exponential distribution? This assumption also seems overly restrictive.

4. Equation 1 should be formulated in terms of empirical loss for given finite sample datasets, rather than using the expectation symbol. This principle should be followed for subsequent equations, such as Equation 2.

5. For Figures 2, 3, and 4, despite their role in elucidating the motivation, the authors do not present evidence of the proposed method's effectiveness in addressing these issues.

6. A number of the methods compared appear to be outdated, predominantly predating 2020, while the most recent ones are from 2021. The authors should consider evaluating against more recent and competitive methods.

**Suitability:**

2

---

### Official Review · Reviewer_gmpo · 2024-05-24

**Rating:** 3
**Confidence:** 4

**Summary:**

The author, through experimental analysis, identified three major drawbacks associated with prior methods for addressing long-tail continuous learning:
1.  The tendency for minority data to be rapidly forgotten.
2.  The propensity for classifier weights to be biased towards the old majority class.
3. The increased likelihood of minority class samples near the decision boundary being forgotten relative to majority class samples in the process of learning following the introduction of new data.

To address these issues, the author introduced the Prior-free Balanced Replay (PBR) framework. This framework enhances the model's memory capacity and mitigates the problem of forgetting. Furthermore, to specifically counter the aforementioned problems, the author proposed the Prototype Constraint module and the Boundary Constraint module. These modules are designed to alleviate the issues of forgetting and bias.

Ultimately, extensive experimental analysis confirmed that the proposed method yields satisfactory results in both ordered- and shuffled-long-tail continuous learning tasks.

**Strengths:**

1. The discourse logic of the article is notably lucid. Firstly, the issue is identified through experimentation, followed by the proposition of a solution for the issue, and finally, the efficacy of the method is substantiated. This organizational structure is exceedingly clear.
2. The Prior-free Balanced Replay method suggested by the author displays a certain degree of innovativeness.
3. When compared to other methodologies, the performance of the method in this article on the ordered-LTCL task is notably enhanced.

**Limitations:**

1. The study asserts in the abstract and introduction that the proposed method exhibits superior performance on ordered-LTCT and shuffled-LTCL tasks. However, the experimental section only provides results related to the shuffled-LTCT task in Table 3, with all other experiments focusing on ordered-LTCT. This does not adequately demonstrate the method's effectiveness on the shuffled-LTCL task.
2. The Prior-free Balanced Replay (PBR) framework, proposed by the author, aims to alleviate the problem of forgetting and enhance the model's performance on older data. In the context of the ordered-LTCT task, the head class is the first to be encountered, meaning that the model's initial interaction is with the highest-performing class. Consequently, one would expect improved performance on the ordered-LTCT task after employing the PBR method. Nevertheless, to substantiate the method's efficacy in mitigating tail class forgetting, the author should conduct ablation and comparative experiments on the shuffled-LTCL task, which would further substantiate the method's effectiveness.
3. The majority of the comparative methods employed in the experiments and the studies followed by this article are from papers published prior to 2022, thus lacking novelty.

**Suitability:**

2

---

### Official Review · Reviewer_1f6x · 2024-05-26

**Rating:** 4
**Confidence:** 4

**Summary:**

This work proposes a novel framework called Prior-free Balanced Replay (PBR) for long-tailed continual learning. It designs an uncertainty-guided reservoir sampling strategy to prioritize rehearsing minority data without using prior information. Meanwhile, PBR includes two other prior-free components to further reduce the forgetting issue including prototype and boundary constraints, which can maintain effective feature information for continually re-estimating task boundaries and prototypes. Extensive experiments on three standard long-tailed datasets demonstrate the superior performance of the proposed PBR over various state-of-the-art methods in both task and class incremental learning settings, as well as ordered- and shuffled-LTCL settings.

**Strengths:**

1.	The elaboration of the paper is clear and reasonable. The authors make it easy to understand the problem of predicting opioid overdose.
2.	The related work does help readers to understand the essence of the idea and to follow the work easily.
3.	The experimental results are well presented and prove that the model is indeed effective.

**Limitations:**

1.	The meaning of Figure 4 is not very clear, why the position near the edge is the task boundary?
2.	The experiment lacks a comparison with the latest methods from 2023 and 2024. I believe that comparing with more recent methods can better highlight the progressiveness of the method proposed in this paper.
3.	In Table 2, ER is better than the proposed method when the buffer size is 500 on the Seq-CIFAR-10-LT (Class-IL). It is necessary to analyze the reason.
4.	If the data is balanced, will the performance of the proposed method also be better than other baselines? Strictly speaking, whether a data is balanced also belongs to prior knowledge. If your method is superior to other baselines even in data balancing, then I believe that your method is indeed not based on prior knowledge. Otherwise, this method is based on prior knowledge.

**Suitability:**

2

---

### Meta-Review · Area_Chair_ZnNQ · 2024-06-30

**Recommendation:** Accept (Poster)
**Confidence:** 5

**Metareview:**

This paper proposes a framework for Long-Tailed continual learning which prioritizes rehearsing minority data without prior information.

In reviews of reviewers, the common concerns raised by most reviewers were insufficient evaluation. In the response, the authors added the latest methods for comparison, adding experiments on a balanced dataset, adding experiments on a shuffled-LTCL task. The method outperformed most of the latest methods except DualHSC, which additionally introduces the task relationships as prior. Currently, Reviewer xh48 has concerns about assumptions. The authors have acknowledged that these assumptions are commonly used in existing works.

The AC believes that major concerns have been reasonably solved, there are only minor concerns about clarification for the assumptions and consistency about problem settings, which can be easily solved in the camera-ready version. Therefore, the AC suggests accepting this paper.

In the camera-ready version, the authors should follow the suggestions from Reviewer xh48 to add a section to clearly discuss the assumptions and limitations caused by the assumptions. Additionally, the paragraph under the problem settings should be revised to follow the settings in the introduction and experiments to avoid confusion.